# Genotypic and Epidemiologic Profiles of *Giardia* *duodenalis* in Four Brazilian Biogeographic Regions

**DOI:** 10.3390/microorganisms10050940

**Published:** 2022-04-30

**Authors:** Deiviane A. Calegar, Beatriz C. Nunes, Kerla J. L. Monteiro, Polyanna A. A. Bacelar, Brenda B. C. Evangelista, Mayron M. Almeida, Jurecir Silva, Jéssica P. Santos, Márcio N. Boia, Lauren H. Jaeger, Filipe A. Carvalho-Costa

**Affiliations:** 1Laboratório de Epidemiologia e Sistemática Molecular, Instituto Oswaldo Cruz, Fundação Oswaldo Cruz, Rio de Janeiro 21045-900, Brazil; polyannabio_gen@hotmail.com (P.A.A.B.); brendaevangelista5@gmail.com (B.B.C.E.); mayronmorais@outlook.com (M.M.A.); jessica.ssantos87@gmail.com (J.P.S.); 2Faculdade de Medicina de Petrópolis (FMP), Centro Universitário Arthur Sá Earp Neto (UNIFASE), Cascatinha, Petrópolis 25600-000, Brazil; biacoronato@gmail.com; 3Escritório Técnico Regional Fiocruz Piauí, Teresina 64000-000, Brazil; kerla.monteiro@gmail.com; 4Instituto Federal de Educação, Ciência e Tecnologia do Piauí, Teresina 64000-000, Brazil; jurecir.silva@ifpi.edu.br; 5Laboratório de Biologia e Parasitologia de Mamíferos Silvestres Reservatórios, Instituto Oswaldo Cruz, Fundação Oswaldo Cruz, Rio de Janeiro 20000-000, Brazil; mboia@ioc.fiocruz.br; 6Faculdade de Farmácia, Universidade Federal de Juiz de Fora, Rua José Lourenço Kelmer, s/n–Campus Universitário, Juiz de Fora 36000-000, Brazil; laurenhj@gmail.com

**Keywords:** *Giardia* *duodenalis*, assemblages, epidemiology, genetic diversity

## Abstract

Human infections with gut protozoan parasites are neglected and not targeted by specific control initiatives, leading to a knowledge gap concerning their regional diversity and epidemiology. The present study aims to explore *Giardia* *duodenalis* genetic diversity and assess the epidemiologic scenario of subclinical infections in different Brazilian biogeographic regions. Cross-sectional surveys (*n* = 1334 subjects) were conducted in four municipalities in order to obtain fecal samples and socioenvironmental data. Microscopy of non-diarrheal feces and nucleotide sequencing of a β-giardin gene fragment were performed. From a total of 51 samples that could be sequenced, 27 (52.9%) β-giardin sequences were characterized as assemblage A and 24 (47.1%) as assemblage B. In the Amazon, assemblage B was the most frequently detected, predominantly BIII, and with two novel sub-assemblages. Assemblage A predominated in the extra-Amazon region, with five novel sub-assemblages. Prevalence reached 17.8% (64/360) in the Amazon, 8.8% (48/544) in the Atlantic Forest, 7.4% (22/299) in Cerrado and 2.3% (3/131) in the Semiarid. People living in poverty and extreme poverty presented significantly higher positivity rates. In conclusion, subclinical giardiasis is endemic in Brazilian communities in different biogeographic regions, presenting high genetic diversity and a heterogeneous genotypic distribution.

## 1. Introduction

*Giardia duodenalis* is a cosmopolitan, flagellated gut protozoan parasite with higher prevalence in developing countries, mainly in regions with poor sanitation and inappropriate drinking water supply [1]. *G. duodenalis* trophozoites inhabit the small intestine and, during the course of infection, a variable proportion of cells encyst and are shed in feces. Cysts can persist in the environment and contaminate water and food, constituting the parasite infective stage. In industrialized regions, *G. duodenalis* causes outbreaks of diarrheal diseases which have water contaminated with fecal material as their main source [2,3].

The Global Enteric Multicenter Study (GEMS) assessed the role of *G. duodenalis* as an etiologic agent of diarrheal diseases in children living in African and Asian developing countries in a 3-year prospective, age-stratified, multicentric and matched case–control study, demonstrating that *G. duodenalis* more frequently infected controls than children aged 12–59 months with diarrhea [4]. Similar results were obtained in Cambodia, where higher *G. duodenalis* positivity rates were found among controls when compared with children with diarrhea [5]. Thus, in endemic areas in developing countries, there is growing evidence that giardiasis is a subclinical and chronic infection, with the parasite being detected in non-diarrheal feces and frequently associated with protein–caloric malnutrition [6,7]. These findings were reported in the Etiology, Risk Factors, and Interactions of Enteric Infections and Malnutrition and the Consequences for Child Health and Development Project (MAL-ED), a multisite birth cohort study [8,9]. Therefore, in vulnerable communities of developing regions, *G. duodenalis*—through complex pathophysiological mechanisms—impacts the absorption of nutrients and affects both the nutritional status and physical development of children in a process that does not depend on the presence of diarrhea [10,11].

*G. duodenalis* infects a wide range of mammalian species and presents considerable intraspecific genetic variation, with eight distinct assemblages (A to H). Assemblages A and B were identified mainly in humans, C and D in wild and domestic canines, E in ruminants and domestic pigs, F in cats, G in mice and rats and H in seals [12,13]. Giardiasis is a potentially zoonotic infection, and cross-host transmission has been well documented [14]. The degree of genetic divergence between assemblages A and B has led to the proposition of two distinct species [15,16]. In Rio de Janeiro, Brazil, assemblage E was detected in human infections [17].

Brazil occupies most of the South American continent and has great biogeographic and climatic diversity, with specific rainfall regimes with great variation, from rain forests to semiarid, associated with different water management strategies, creating regional scenarios for the epidemiology of water-borne infections.

Infections with gut protozoan parasites are not targeted by specific intestinal parasite control strategies, which are based on the mass administration of anthelmintic drugs [18]. This has led to a knowledge gap concerning the prevalence, distribution and factors associated with subclinical giardiasis in many regions, as well as concerning parasite genetic diversity. The present study aims to describe the parasite genetic heterogeneity and the epidemiological scenario of subclinical *G. duodenalis* infection in an urban Amazonian area and three extra-Amazonian Brazilian biogeographic regions.

## 2. Materials and Methods

### 2.1. Study Design

Cross-sectional surveys (*n* = 1334; Table 1) were performed in order to obtain sociodemographic, anthropometric and sanitation data as well as fecal samples for parasitological and molecular analyses (Figure 1). For logistical reasons in the field work, only one fecal sample was obtained from each person. The interviews were conducted face-to-face by the research team in domiciles. Fecal samples were from non-diarrheal stools, from asymptomatic subjects. Weight (*n* = 844) and height (*n* = 742) were obtained from individuals between 0–14 years of age. Weight was obtained with a portable electronic scale, to the nearest 100 g. Children were barefoot and with minimum clothing. Infants aged less than 12 months were weighed in their mothers’ arms. Height or length was measured using an anthropometer to the nearest 0.1 cm. The nutritional parameters height-for-age Z-scores (HAZ) and weight-for-age Z-scores (WAZ) were calculated with the Nutrition module of the Epi Info™ v.3.5.1 software to verify the presence of protein–energy malnutrition characterized by stunting (HAZ < −2) and low weight (WAZ < −2). Extreme poverty was defined when monthly family per capita income was below BRL 125, which corresponds to USD 25 (considering the exchange rate of USD 1 = BRL 5). Poverty was defined by monthly family per capita income between BRL 125 and BRL 250 (USD 25–50). The researchers gathered information about the site of defecation, i.e., if the family had a latrine inside the house or if members of the family practice open defecation in the peridomestic environment. The final destination of the feces was also characterized, being adequate when the feces went to closed septic tanks and inadequate when they were deposited in the ground, in rudimentary holes or directly into a waterbody.

### 2.2. Socio-Environmental Characteristics of the Biogeographic Regions

Brazil has great social and environmental diversity. Many regions have poor access to drinking water and inadequate sewage systems in common. The study was conducted in four localities whose sociodemographic, environmental and climatic differences are presented in Table 1: São João do Piauí and Teresina, in the state of Piauí; Bagre, in the state of Pará; and Cachoeiras de Macacu, in the state of Rio de Janeiro (see map in Figure 2).

### 2.3. Parasitological Examinations

Fecal samples were collected in plastic bottles without preservatives and sent to the field laboratory to be examined through light microscopy using the Ritchie and saturated glucose solution flotation techniques. For the Ritchie method, fecal suspensions were homogenized, filtered through folded gauze and centrifuged at 2500 rpm for 1 min. Sediment was resuspended with neutral detergent and water and centrifuged at 2500 rpm for 1 min. Sediment was examined through light microscopy. For the saturated glucose solution flotation technique, fecal material subjected to a hyperosmolar sucrose solution was collected on the surface with the aid of a loop and examined on a microscope slide. Two experienced parasitologists examined all samples.

### 2.4. DNA Extraction, PCR Amplification and Nucleotide Sequencing of Β-Giardin Encoding Gene Fragment

After parasitological examinations, 137 *G. duodenalis*-positive fecal samples were obtained. Since the success rate in the amplification and sequencing of parasitological material of fecal origin is usually low, and to enable a greater number of sequences, PCR of a pool of microscopically negative samples was also performed (*n* = 42), usually from inhabitants who lived in the homes of positive subjects. Thus, a total of 179 fecal samples were submitted for PCR. Genomic DNA was extracted from 200 µL of the sedimented fecal material using the ZR Fungal/Bacterial DNA MiniPrep™ kit (ZymoResearch, Irvine, CA, USA). PCR was performed using the Platinum Taq DNA Polymerase kit (Invitrogen, Waltham, MA, USA) with a final volume of 50 μL, and targeted a 753 bp region of the β-giardin locus of *G. duodenalis*, as described [19]. The PCR conditions were: 1X PCR Buffer, 1.5 mM MgCl_2_, 0.05 mM dNTP, 10 pmol of each primer, 2.5 U of Taq polymerase and ~40 ng of template DNA. Amplification parameters included an initial denaturation at 94 °C for 5 min followed by 35 cycles of amplification comprising denaturation (94 °C for 30 s), annealing (65 °C for 30 s) and extension (72 °C for 30 s), and a final extension at 72 °C for 5 min. The PCR products were purified with polyethylene glycol (PEG). Capillary electrophoresis was performed in an ABI3730 automated DNA sequencer (Applied Biosystems) in PDTIS/Fiocruz Genomic Platform RPT01A.

### 2.5. Sequence Data Analysis

The sequences were edited and analyzed using the BioEdit v.7.2.5 software [20]. The Basic Local Alignment Search Tool (BLAST–NCBI https://www.ncbi.nlm.nih.gov accessed on 20 August 2021) was used to verify similarity with *G. duodenalis* sequences. All sequences generated were deposited in the GenBank database under the accession numbers MW679411-MW679461. To determine *G. duodenalis* assemblages (genotypes), an alignment was performed with 55 *G. duodenalis* orthologous reference sequences retrieved from GenBank in BioEdit v.7.2.5 software. Sequences with degenerate bases were not included. Further details of reference strains can be found in Appendix A. The most appropriate substitution model was estimated using Bayesian Information Criterion (BIC) in MEGA v.7 software [21]. Maximum likelihood (ML) and neighbor joining (NJ) genetic trees were constructed in MEGA v.7 software using a Tamura-Nei model (bootstrap 1000-replicates). The median-joining (MJ) haplotype network based on distance criteria was constructed using the Network v.10.1.0.0 software (Fluxus Technology Ltd., www.fluxusengineering.com) [22]. The DNA Sequence Polymorphism (DnaSP) v.5.10.01 software was used for editing the files [23]. To evaluate the intraspecific genetic diversity of *G. duodenalis*, diversity indexes were determined for each population pair using Arlequin v.3.5.2.2 software (http://cmpg.unibe.ch/software/arlequin35 accessed on 20 August 2021) [24]. The populations were grouped considering assemblage, geographic origin and Brazilian regions.

### 2.6. Statistical Analysis

*G. duodenalis* positivity rates were described in different groups defined by sociodemographic characteristics and nutritional status. Prevalence ratios (PRs) and their respective 95% confidence intervals (CIs) were calculated. The statistical significance of the differences between the positivity rates was assessed by Fisher’s exact test. Associations were considered statistically significant when *p* < 0.05. Statistical analyses were performed with Epi Info 2000^®^ (CDC, Atlanta, GA, USA).

### 2.7. Ethics

The study was approved by the Research Ethics Committee (license CAAE 12125713.5.0000.5248) of the Oswaldo Cruz Institute, Fiocruz.

## 3. Results

### 3.1. Genetic Diversity of Giardia duodenalis

Of the 179 fecal samples submitted to PCR/sequencing, 51 were successfully genotyped using the β-giardin locus. Overlapping peaks were not observed in the nucleotide sequences. Twenty-seven (52.9%) sequences were characterized as assemblage A and twenty-four (47.1%) as assemblage B (Table 2).

The MJ haplotype network (Figure 3) showed that the *G. duodenalis* sequences were grouped by assemblages, as expected. Assemblages A and B presented a star-like shape, including the sequences obtained in this study as a central and dominant haplotype (except the novel haplotypes).

ML and NJ phylogenetic trees (Figure 4) also demonstrated that the *G. duodenalis* sequences were grouped by assemblages. The main difference between the two trees was in the NJ tree: assemblage D shared a common ancestor with assemblages B and E. Concerning the molecular diversity indexes, in general, assemblage B revealed greater intraspecific diversity when compared with assemblage A (H = 0.921 ± 0.028 vs H = 0.854 ± 0.029) (Appendix A). In Brazil, assemblage A showed greater intraspecific diversity (H = 0.879 ± 0.037) when compared with Europe and North America (H = 0.822 ± 0.096 and 0.666 ± 0.204, respectively). In contrast, assemblage B in Brazil showed lower intraspecific diversity when compared with North America, Asia and Europe (H = 0.918 ± 0.033 vs. 1.000 ± 0.500, 1.000 ± 0.126 and 1.000 ± 0.272). Assemblage A from São João do Piauí showed lower diversity when compared with the other biomes in the present study. Assemblage B from the Amazon biome showed a lower diversity (H = 0.757 ± 0.086) when compared with reference sequences from the same biome (H = 0.991 ± 0.025) (Appendix A).

### 3.2. Prevalence and Factors Associated with Giardia duodenalis Infection

Table 3 shows positivity rates (by microscopy) in different groups defined by sociodemographic characteristics and nutritional status and the association of giardiasis with other intestinal protozoa (coinfection with *Entamoeba histolytica/E. dispar* and with *Entamoeba coli*). The overall positivity rate was 137/1334 (10.3%). The frequency was significantly higher in Bagre, in the Amazon region, reaching 64/360 (17.8%) and lower in São João do Piauí, in the Caatinga (3/131 (2.3%)). In Bagre, Teresina and Cachoeiras de Macacu, the age groups of 3 to 6 years old and 7 to 15 years old had the highest rates of positivity, but infants and toddlers up to 2 years old were also frequently infected in Bagre and Cachoeiras de Macacu. Giardiasis was significantly more frequent among people living in poor and extremely poor families in Bagre, Teresina and Cachoeiras de Macacu. People living in scenarios of inappropriate disposal of feces and open defecation also presented significantly higher positivity. Individuals positive for *Entamoeba histolytica*/*E. dispar* or *E. coli* were infected with *G. duodenalis* at significantly higher frequencies.

## 4. Discussion

The present study explored regional differences in the genotypic composition and epidemiologic profile of *G. duodenalis* in different Brazilian biogeographic regions. Regarding the detection frequencies of the main assemblages—even considering the relatively small proportion of positive samples in which the partial sequencing of the β-giardin gene was achieved—some regional differences could be observed. It is important to note that the large amount of bacterial DNA in the fecal samples, as well as inhibitory substances in the material, usually interfered negatively in the success of PCR amplification.

In the Amazon region, there was a predominance of assemblage B, and in the extra-Amazonian area, assemblage A was more frequently detected. A similar genotypic profile was described in a previous work when we compared another Amazon region (the Rio Negro basin) with different extra-Amazonian regions [25,26]. A high detection rate of assemblage B was also detected in the Colombian Amazon [27]. In the present study, besides differences in genotypic composition, the Amazon region presented a substantially higher prevalence of *G. duodenalis* subclinical infection. In São Paulo (southeastern Brazil), a predominance of assemblage A was reported in low-income families [28]; this predominance was also demonstrated in environmental samples recovered from sewage in the same region [29]. In Rio de Janeiro, an increase in the proportion of infections by assemblage B in recent years has been suggested, with this genotype being associated with HIV coinfection and more severe symptoms, influenced by the degree of immunosuppression presented by patients [30,31]. A prospective study carried out in Fortaleza, northeastern Brazil, suggests that the parasite burden of *G. duodenalis* assemblage B infections is higher, with greater shedding of cysts and, consequently, a greater potential for spreading [32]. Outside of the Amazon, the predominance of assemblage A was also demonstrated in southern and northeastern Brazil [33,34].

Taken together, these data may point to putative differences in the epidemiologic profile of *G. duodenalis* assemblages A and B. However, prospective studies with large sample sizes and quantitative assessments of the parasite load, response to treatment and the clinical and nutritional impact of giardiasis have not yet been carried out in order to assess clear clinical and epidemiological differences between the main genotypes of *G. duodenalis*. The present study did not include subjects with diarrhea, and infections were identified in subclinical conditions. In Saudi Arabia, children infected with assemblage B were predominantly symptomatic, whereas asymptomatic participants harbored assemblages AI and AII [35]. Conversely, in Egypt, it was demonstrated that iron deficiency anemia and intestinal symptoms were mainly associated with assemblage A [36].

Despite the lack of robust data on the clinical and epidemiological differences between *G. duodenalis* assemblages A and B, the genetic variation between them is well established. *G. duodenalis* complete genomes revealed substantial phylogenetic divergence between the two main genotypes infecting humans and demonstrate that: (i) the average amino-acid identity in 4300 orthologous proteins is not superior to 78% and (ii) the full-genome-derived similarity between enzootic assemblage E and assemblages A and B at the level of amino-acid identity is 90% and 81%, respectively [31]. Our phylogenetic inferences based on the β-giardin gene partial sequencing also demonstrated marked divergence between assemblages A and B in Brazil. Through the ML analysis, assemblage E sequences obtained in GenBank were more related to our samples characterized as assemblage A, while the NJ analysis demonstrated that assemblage E was closer to assemblage B. In both phylogenetic trees, assemblage A was more closely related to enzootic assemblages C and F than to assemblage B.

The comparison of some socio-environmental characteristics of the four regions assessed in the present study reveals that Bagre in the Amazon region, the locality with higher prevalence of *G. duodenalis* infection, has the lowest human development index and the highest proportion of people living in poverty and extreme poverty. Historically, in the Amazon, the process of demographic concentration of populations of Amerindian descent has favored the spread of parasitic and fecal-borne diseases. The Amazon is the region of the country with the highest prevalence of soil-transmitted helminthiases, incidence of diarrheal diseases and diarrhea-related mortality. The riverside character of Bagre should also be considered, with its close proximity and greater contact of the population with waterbodies in a region of abundant rainfall and poor sanitation, favoring the spread of water-borne infections, which contrasts with the semiarid climate and very low rainfall in São João do Piauí, where the prevalence of giardiasis was lower. Intermediate positivity rates were observed in Teresina and Cachoeiras de Macacu, in the Cerrado and Atlantic Forest biogeographic regions, respectively. In addition, our data demonstrated that *G. duodenalis* positivity is strongly influenced by the living conditions of the studied subjects, as the positivity rate was higher among children within the poorest families in Bagre, Teresina and Cachoeiras de Macacu and with inadequate destination of feces in Teresina. The detected association of giardiasis with other gut protozoan parasites, such as *E. histolytica*/*E. dispar* and *Entamoeba coli,* reinforces the vulnerability of the poorest families to fecal-borne infections.

In conclusion, subclinical giardiasis, identified in subjects without diarrhea, is frequent in resource-poor Brazilian communities and with a heterogeneous geographic genotype distribution and prevalence. The data suggest the need to improve control strategies, including better access to diagnosis and treatment.

## Figures and Tables

**Figure 1 microorganisms-10-00940-f001:**
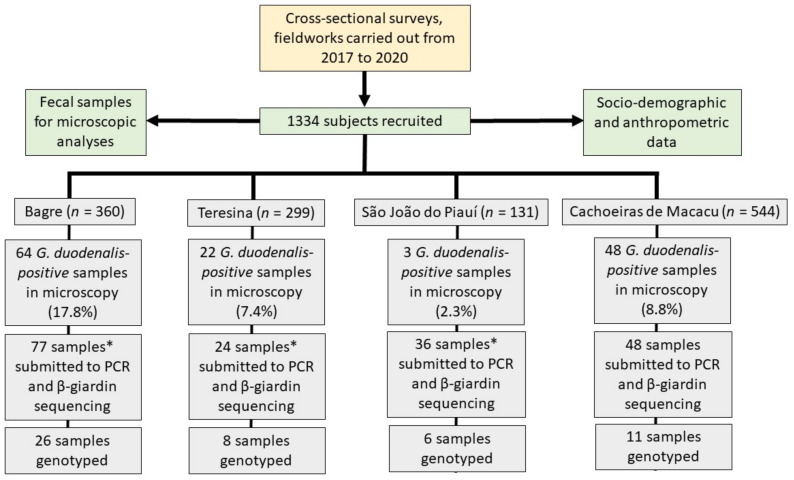
Flowchart illustrating the study design and geographic distribution of fecal samples analyzed, including those positive for *Giardia duodenalis* and those that could be genotyped by partial sequencing of the giardin beta gene. * Includes negative samples on microscopy, randomly or when another individual from the same household was positive with the following distribution: 7 in Bagre, 2 in Teresina and 33 in São João do Piauí.

**Figure 2 microorganisms-10-00940-f002:**
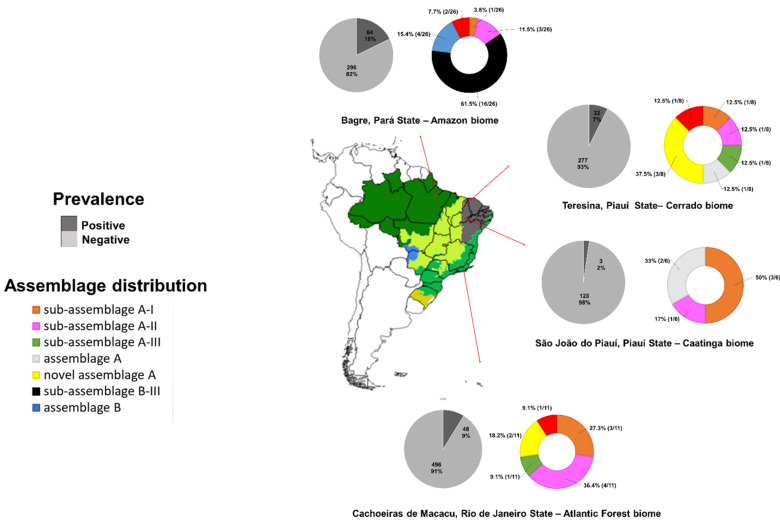
Geographical distribution of the different assemblages and sub-assemblages of *Giardia duodenalis* in the studied localities.

**Figure 3 microorganisms-10-00940-f003:**
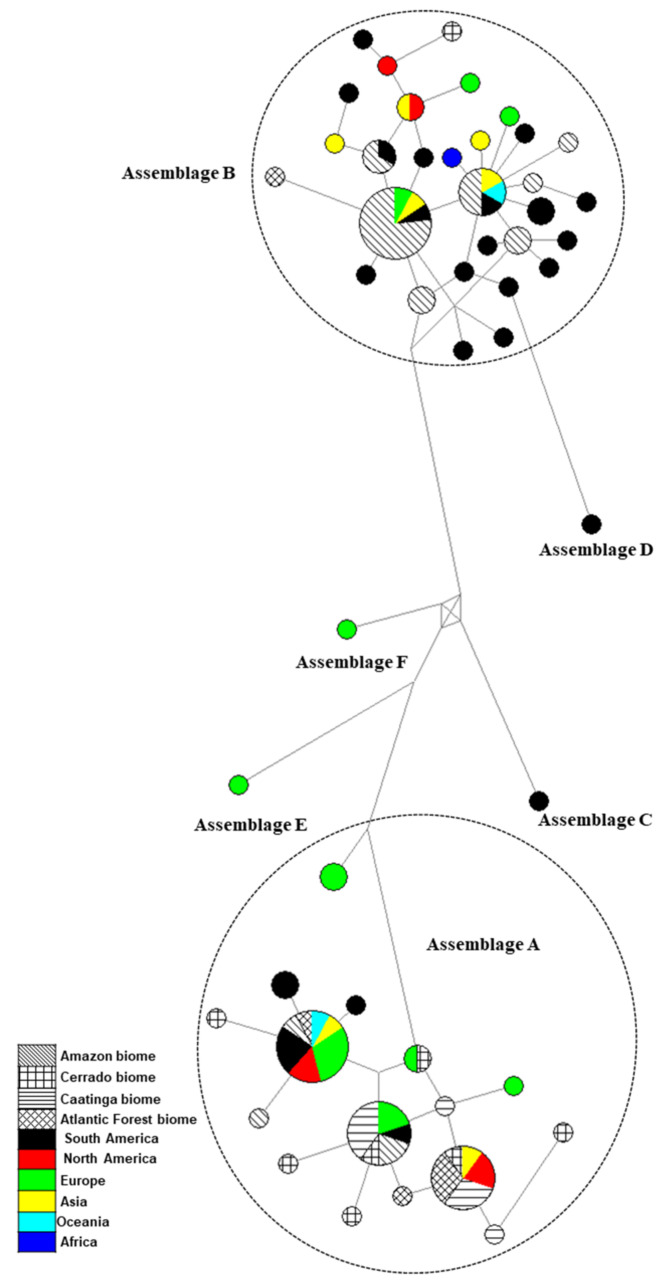
Haplotype network based on *Giardia duodenalis* β-giardin locus (592 bp, *n* = 106). Area of the circle is proportional to number of sequences. Further details of reference strains can be found in Appendix A.

**Figure 4 microorganisms-10-00940-f004:**
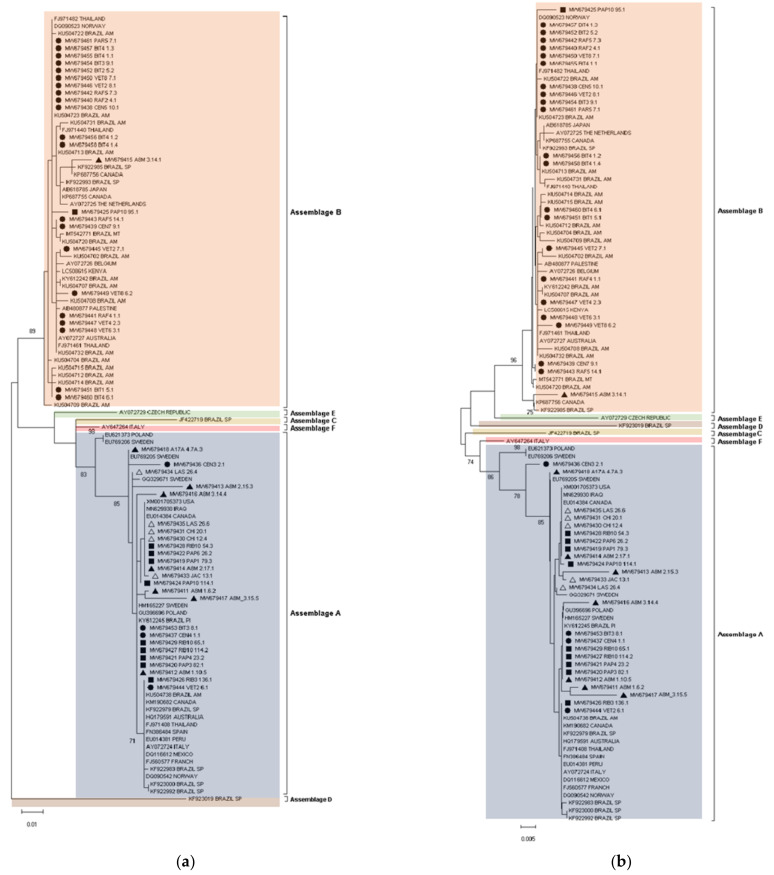
(**a**) Maximum likelihood and (**b**) neighbor joining trees inferred from *Giardia duodenalis* β-giardin locus (592 bp, *n* = 106). Support for the branching order was determined by 1000 bootstrap replicates, and only values >70% are reported. ●: Bagre—Amazon; ▪: Cachoeiras de Macacu—Atlantic Forest; ∆: São João do Piauí—Caatinga; ▲: Teresina—Cerrado biome. GenBank accession numbers are indicated. Further details of reference strains can be found in Appendix A.

**Table 1 microorganisms-10-00940-t001:** Socio-environmental characteristics of the different studied localities.

Characteristic	Municipality/Biogeographic Region/State			
	Bagre/Amazon/PA	Teresina/Cerrado/PI	São João do Piauí/Caatinga/PI	Cachoeiras de Macacu/Atlantic Forest/RJ
**Population**	30,673	864,845	20,601	58,937
**Climate ***	Equatorial (Af)	Semi-humid tropical (Aw)	Semiarid (Bsh)	Semi-humid tropical (Aw)
**Income (MPCHI) ****				
**Extreme poverty ^a^**	144/360 (40%)	107/299 (35.8%)	44/131 (33.6%)	159/544 (29.2%)
**Poverty ^b^**	114/360 (31.7%)	91/299 (30.4%)	52/131 (39.7%)	113/544 (20.8%)
**Human development index**	0.471	0.751	0.645	0.700
**Gini index**	0.37	0.50	0.45	0.45
**Water supply**	Furo de Santa Maria River (Baía de Marajó)	Artesian wells	Rain water stored in cisterns and artesian wells	Macacu River and artesian wells
**Year of study**	2020	2017	2018	2018
**Localization of districts** **included**	Urban	Rural	Rural	Urban and rural
**Fecal samples**	360	299	131	544

AM: Amazonas State, PA: Pará State, PI: Piauí State, RJ: Rio de Janeiro State. * World map of the Koppen–Geiger climate classification system. ** MPCHI—monthly per capita house income (1/4 of minimum wage, considering BRL 1000). ^a^ MPCHI < BRL 125 per capita. ^b^ MPCHI = BRL 125–250 per capita.

**Table 2 microorganisms-10-00940-t002:** Distribution of *Giardia duodenalis* assemblages and sub-assemblages obtained in this study based on β-giardin locus (592 bp, *n* = 51).

Assemblage/Sub-Assemblage	Localities				Total
	Bagre	Cachoeiras de Macacu	São João do Piauí	Teresina	
A	-	-	2	1	3
AI	1	3	3	1	8
AII	3	4	1	1	9
AIII	-	1	-	1	2
A novel	-	2	-	3	5
B	4	-	-	-	4
BIII	16	-	-	-	16
B novel	2	1	-	1	4
**Total**	**26**	**11**	**6**	**8**	**51**

**Table 3 microorganisms-10-00940-t003:** Positivity rates in different groups defined by sociodemographic characteristics and the association of giardiasis with other intestinal protozoa (*Entamoeba histolytica/Entamoeba dispar* or *Entamoeba coli*) in different Brazilian biogeographic regions.

	Bagre/Amazon/Pará				Teresina/Cerrado/Piauí				São João do Piauí/Caatinga/Piauí				Cachoeiras de Macacu/Atlantic Forest/RJ			
	Positivity (%)	PR	95% CI	p	Positivity (%)	PR	95% CI	p	Positivity (%)	PR	95% CI	p	Positivity (%)	PR	95% CI	p
Sex																
Male	34/186 (18.3)	1.07	0.62–1.84	0.797	11/147 (7.5)	1.03	0.46–2.31	0.935	2/71 (2.8)	1.69	0.15–18.18	0.662	25/284 (8.8)	0.99	0.57–1.70	0.985
Female	30/174 (17.2)	1			11/152 (7.2)	1			1/60 (1.7)	1			23/260 (8.8)	1		
**Age group**																
0–2	8/77 (10.4)	1		0.027	0/11 (0)	-			1/7 (14.3)	1			3/99 (3.0)	1		
3–6	20/133 (17.7)	1.56	0.63–3.65	0.340	2/21 (9.5)	1			1/8 (12.5)	1.14	0.08–15.07	0.921	19/164 (11.6)	3.82	1.16–12.59	0.015
7–15	36/170 (21.2)	2.03	0.99–4.17	0.040	6/58 (10.3)	1.08	0.23–4.96	0.915	1/27 (3.7)	0.29	0.02–4.22	0.353	22/234 (9.4)	3.10	0.95–10.12	0.044
16–30	-				2/44 (4.5)	0.47	0.07–3.15	0.438	0/34 (0)	-			0/1 (0)	-		
31–50	-				5/84 (6)	0.62	0.13–2.99	0.559	0/32 (0)	-			1/1 (100)	-		
>50	-				0/1 (0)	-			0/22 (0)	-			0/1 (0)	-		
**Income**																
>250	7/100 (7)	1			2/101 (2)	1			0/35 (0)	-			13/209 (6.2)	1		
125-250	23/114 (20.2)	2.88	1.29–6.42	0.006	8/91 (8.8)	4.43	0.96–20.36	0.034	2/52 (3.8)	1			11/113 (9.7)	1.56	0.72–3.37	0.252
<125	34/144 (23.6)	3.37	1.55–7.30	<0.001	12/107 (11.2)	5.66	1.29–24.68	0.008	1/44 (2.3)	0.59	0.05–6.30	0.660	19/159 (11.9)	1.92	0.97–3.77	0.053
**Open defecation**																
Yes	21/103 (20.4)	1.21	0.75–1.93	0.430	13/106 (12.3)	2.63	1.16–5.94	0.016	2/87 (2.3)				0/2 (0)	-		
No	43/255 (16.9)	1			9/193 (4.7)	1			1/44 (2.3)	1			43/477 (9)	-		
**Inadequate feces disposal**																
Yes	64/360 (17.8)				17/153 (11.1)	3.24	1.22–8.56	0.011	2/87 (2.3)	1.01	0.09–10.85	0.993	24/271 (8.9)			
No	-				5/146 (3.4)	1			1/44 (2.3)	1			15/182 (8.2)			
**Stunting**																
Yes	12/51 (23.5)	1.35	0.76–2.37	0.305	1/6 (16.7)	1.61	0.23–11.06	0.632	-	-			1/16 (6.3)	0.60	0.08–4.15	0.602
No	43/247 (17.4)	1			7/68 (10.3)	1			-	-			39/380 (10.3)	1		
**Low weight**																
Yes	7/47 (14.9)	0.81	0.39–1.67	0.564	1/4 (25)	2.50	0.39–15.69	0.350	-	-			3/21 (14.3)	1.56	0.52–4.64	0.429
No	54/294 (18.4)	1			7/70 (10)	1			-	-			40/438 (9.1)	1		
** *Entamoeba coli coinfection* **																
Yes	26/80 (32.5)	2.39	1.55–3.69	<0.001	6/59 (10.2)	1.52	0.62–3.72	0.356	1/32 (3.1)	1.54	0.14–16.50	0.717	6/30 (20)	2.44	1.13–5.29	0.026
No	38/280 (13.6)	1			16/240 (6.7)	1			2/99 (2.0)	1			42/514 (8.2)	1		
** *Entamoeba histolytica/ E. dispar coinfection* **																
Yes	26/53 (49.1)	3.96	2.64–5.94	<0.001	9/28 (32.1)	6.70	3.14–14.26	<0.001	0/1 (0)	-			21/71 (29.6)	5.18	3.10–8.65	<0.001
No	38/307 (12.4)	1			13/271 (4.8)	1			3/130 (2.3)	-			27/473 (5.7)	1		

## Data Availability

All nucleotide sequencing data have been submitted to GenBank as shown in the methods section. The data that support the findings of this study are available on request from the corresponding author, Carvalho-Costa FA. The data are not publicly available because they contain information that could compromise the privacy of research participants.

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
