# Peer review of "Genotypic and Epidemiologic Profiles of Giardia duodenalis in Four Brazilian Biogeographic Regions"

_microorganisms, 2022, doi:10.3390/microorganisms10050940_

Round 1
Reviewer 1 Report
This valuable study reports genetic diversity and epidemiology of Giardia intestinalis in Brazilian people. This study is well constructed and written. Mixed uses of full names and abbreviated names of localities complicate the readers probably. Just the reviewer suggests some editorial points for revision.
L23: ‘Intestinal protozoan parasites in humans’ (?), or generally indicated?
L32, L218, L271: ‘Prevalence’ instead of ‘Prevalence rate’.
L53: ‘Cambodia’ in English, please replace ‘Cambodja’ by ‘Cambodia’.
L94: Resolution of Figure 1 is poor. Replace it by one with high resolution.
L133, L148, L212: ‘G. duodenalis’ in italic.
L134: ‘MgCl2’; subscribe ‘2’ after ‘Cl’.
L166: ‘p’ in italic, which appears prior to ‘<0.05’.
L167: ‘2000®’; superscript ‘®’ after ‘2000’.
L180, L182: ‘22/26 (84.6%) ? ‘4 samples (15.4%)’ ?
L181: Delete ‘de’ between ‘with’ and ‘following’.
L182-183: ‘AI, n=1 and AII, n=3’ ?
Fig 2: ‘Prevalence’ instead of ‘Prevalente rate’
L192: AI in all sites including Amazon biome, as shown in Table 2.
L234-235: ‘Entamoeba histolytica’, ‘Entamoeba dispar’, and ‘Entamoeba coli’ in italic.
L239: Delete ‘associated’, or ‘socioenvironment-associated’.
L274: ‘fecal’
L327-L405: Use an identical format for all references (Change unnecessary capital letters into small letters). Scientific names in italic (Please check each original publication).
Author Response
The authors thank the corrections and inform that they were all incorporated into this new version of the manuscript
L23: ‘Intestinal protozoan parasites in humans’ (?), or generally indicated?
Corrected
L32, L218, L271: ‘Prevalence’ instead of ‘Prevalence rate’.
Corrected
L53: ‘Cambodia’ in English, please replace ‘Cambodja’ by ‘Cambodia’.
Corrected
L94: Resolution of Figure 1 is poor. Replace it by one with high resolution.
We improved Figure resolution
L133, L148, L212: ‘G. duodenalis’ in italic.
Corrected
L134: ‘MgCl2’; subscribe ‘2’ after ‘Cl’.
Corrected
L166: ‘p’ in italic, which appears prior to ‘<0.05’.
Corrected
L167: ‘2000®’; superscript ‘®’ after ‘2000’.
Corrected
L180, L182: ‘22/26 (84.6%) ? ‘4 samples (15.4%)’ ?
Corrected
L181: Delete ‘de’ between ‘with’ and ‘following’.
Corrected
L182-183: ‘AI, n=1 and AII, n=3’ ?
Corrected
Fig 2: ‘Prevalence’ instead of ‘Prevalente rate’
Corrected
L192: AI in all sites including Amazon biome, as shown in Table 2.
L234-235: ‘Entamoeba histolytica’, ‘Entamoeba dispar’, and ‘Entamoeba coli’ in italic.
Corrected
L239: Delete ‘associated’, or ‘socioenvironment-associated’.
Corrected
L274: ‘fecal’
Corrected
L327-L405: Use an identical format for all references (Change unnecessary capital letters into small letters). Scientific names in italic (Please check each original publication
We adjusted all references
Reviewer 2 Report
The manuscript “Genotypic and epidemiologic profiles of Giardia duodenalis in 2 four Brazilian biogeographic regions” aims to explore Giardia duodenalis genetic diversity and assess the epidemiologic scenario of subclinical infections in different Brazilian biogeographic regions. It is an important study area using adequate methodologies; however, the manuscript seems to be the forced joining of two different studies, namely an epidemiological study involving 1334 subjects and 4 sampling sites across Brazil, including 4 different biomes, and a second study involving genetic analysis of just 51 samples with different representativity of each biome. (I would consider that the sample size used for genetic analysis is not enough to support the conclusion of the manuscript.
The aim of the project is well defined, the problem is important however the manuscript is not well organized, several edit errors. If possible, I would suggest increasing the genetic analysis data (at least 10% from each site, meaning study about 140 samples). Moreover, a reorganization of the manuscript, including limitations of the study is needed.
Specific suggestions
Abstract Line 28 the number of samples used in genetic analysis is not mentioned
Introduction – line 72 to 74. It is confused, because authors mentioned that in Brazil assemblage B is presented in less extent, but in line 74 it is predominant in Amazon, that is also Brazil
Material and methods:
Study design – Line 101 – authors use just one fecal sample, and they do not discuss the potential impact of just one sampling in the prevalence of Giardia. (Several studies use 3 samples)
Line 103 and 104- No information concerning equipment used for anthropometric data or even of it was self-reported. Moreover, the number of children in this age classes is not mentioned.
Throughout the all document the name of the sampling sites is not uniform creating difficulties to follow, for instance in table 1 is Amazon/Bagre (PA), in table 2 is Bagre (AM), in line 180 is Amazon (BAG) in figure 2 is Bagre, Pará State
2.3 Parasitological examination – Authors need to describe the method and how many microscopist have observed each sample.
Results –
I think this is the most problematic issue of the manuscript. Authors have used 180 samples for genetic analysis (137 positive and 43 negatives from microscopy) and just have positive results in 51. No information concerning the possible causes of unsuccess, there is no information concerning the age or the place if sample from those samples (try to prove that the missing ones does not influence the results). Also, there is no information concerning sensibility or specificity of the microscopists. Finally, as I have already mentioned the sample size (51) id very short to get conclusions as claimed by the authors.
Line 180 – the number of samples in assemblage B is 22/26 (not 28) being a prevalence of 84.6%
Line 181 to 183 – repeated from table 2
Line 187 – 195 – the majority is a repletion from table 2 (in line 190 is referred table 3 but should be table 2)
Figure 2 – The results from Bagre are not correct. The sum is 27, but the number in table 2 is 26. In the figure there is a reference to Assemblage A (1 sample) but in table 2 there is no such sample.
Figure 3 . It could be important to reference the origin of the samples analyzed (as in figure 4).
Line 208 – Where is Caatinga (as mentioned it is difficult to follow the locations because of different nomenclature and names used)
Table 3 – There is no need to include the “Unknown”
Table 3 – results. It would be interesting in presenting table 3 results to refer if location influences the results. For instance, if the high prevalence of positivity in one place is associated with the worst sanitary conditions in that place. I would suggest these correlations to be performed by site and not in the total sample. The sample size for the cross-sectional study is enough to perform this analysis
Discussion. Should be reorganized because it is very confusing.
Lines 256-259 – All this paragraph could be deleted because it is very difficult to read and doesn’t give ny new information-
There is no reference to the small sample size used in the genetic analysis
There is no limitations of the study
Author Response
The aim of the project is well defined, the problem is important however the manuscript is not well organized, several edit errors. If possible, I would suggest increasing the genetic analysis data (at least 10% from each site, meaning study about 140 samples). Moreover, a reorganization of the manuscript, including limitations of the study is needed.
Answer: We agree that there are several editing errors and have restructured the manuscript. We rewrote the discussion, deleted a figure and made a new table with the prevalence results. All required corrections below have been made, which has greatly improved the work. We thank you for the thorough review. In the discussion, we added a specific paragraph about limitations. However, we could not increase the proportion of positive samples that could be genotyped. In fact, the success rate on genotyping of Giardia duodenalis recovered from feces is usually low, mainly due to the large amount of bacterial DNA in the material, which negatively impacts PCR amplification. Recently, in a study carried out in Colombia by Higuera et al., it was possible to genotype 33 samples by gdh and 25 samples by tpi squencing in a universe of 280 fecal samples positive for Giardia duodenalis (proportion of genotyped isolates = 58/280; 20%) (Higuera A, Villamizar X, Herrera G, Giraldo JC, Vasquez-A LR, Urbano P, Villalobos O, Tovar C, Ramírez JD. Molecular detection and genotyping of intestinal protozoa from different biogeographical regions of Colombia. PeerJ. 2020 Mar 9;8:e8554). We have added a paragraph in the Discussion explaining the reason for the relatively low success rate in amplification/sequencing of the chosen genetic target. Regarding the apparent dissociation between the epidemiological survey and the molecular characterization, we tried to improve the link between both, mainly through a flowchart, which was added to the work.
Specific suggestions
Abstract Line 28 the number of samples used in genetic analysis is not mentioned
Answer: We corrected it.
Introduction – line 72 to 74. It is confused, because authors mentioned that in Brazil assemblage B is presented in less extent, but in line 74 it is predominant in Amazon, that is also Brazil
Answer: We corrected it.
Material and methods:
Study design – Line 101 – authors use just one fecal sample, and they do not discuss the potential impact of just one sampling in the prevalence of Giardia. (Several studies use 3 samples)
Answer: We added information about it.
Line 103 and 104- No information concerning equipment used for anthropometric data or even of it was self-reported. Moreover, the number of children in this age classes is not mentioned.
Answer: We added information about it.
Throughout the all document the name of the sampling sites is not uniform creating difficulties to follow, for instance in table 1 is Amazon/Bagre (PA), in table 2 is Bagre (AM), in line 180 is Amazon (BAG) in figure 2 is Bagre, Pará State
Answer: We adjusted it.
2.3 Parasitological examination – Authors need to describe the method and how many microscopist have observed each sample.
Answer: We added information about it.
Results –
I think this is the most problematic issue of the manuscript. Authors have used 180 samples for genetic analysis (137 positive and 43 negatives from microscopy) and just have positive results in 51. No information concerning the possible causes of unsuccess, there is no information concerning the age or the place if sample from those samples (try to prove that the missing ones does not influence the results). Also, there is no information concerning sensibility or specificity of the microscopists. Finally, as I have already mentioned the sample size (51) id very short to get conclusions as claimed by the authors.
Answer: We added information on sample loss, i.e., the proportion that could not be genotyped.
Line 180 – the number of samples in assemblage B is 22/26 (not 28) being a prevalence of 84.6%
Answer: We adjusted it.
Line 181 to 183 – repeated from table 2
Answer: We adjusted it.
Line 187 – 195 – the majority is a repletion from table 2 (in line 190 is referred table 3 but should be table 2)
Answer: We adjusted it.
Figure 2 – The results from Bagre are not correct. The sum is 27, but the number in table 2 is 26. In the figure there is a reference to Assemblage A (1 sample) but in table 2 there is no such sample.
Answer: We adjusted it.
Figure 3 . It could be important to reference the origin of the samples analyzed (as in figure 4).
Answer: We adjusted it.
Line 208 – Where is Caatinga (as mentioned it is difficult to follow the locations because of different nomenclature and names used)
Answer: We corrected it.
Table 3 – There is no need to include the “Unknown”
Answer: We adjusted it.
Table 3 – results. It would be interesting in presenting table 3 results to refer if location influences the results. For instance, if the high prevalence of positivity in one place is associated with the worst sanitary conditions in that place. I would suggest these correlations to be performed by site and not in the total sample. The sample size for the cross-sectional study is enough to perform this analysis
Answer: We have replaced this table with a new one, with analysis separated by location. We agreed that the previous table did not allow for correct interpretations of factors associated with giardiasis.
Discussion. Should be reorganized because it is very confusing.
Lines 256-259 – All this paragraph could be deleted because it is very difficult to read and doesn’t give ny new information-
There is no reference to the small sample size used in the genetic analysis
There is no limitations of the study
The aim of the project is well defined, the problem is important however the manuscript is not well organized, several edit errors. If possible, I would suggest increasing the genetic analysis data (at least 10% from each site, meaning study about 140 samples). Moreover, a reorganization of the manuscript, including limitations of the study is needed.
Answer: We rewrote the entire Discussion, limiting ourselves to discussing what the data actually allows, within the limitations of sample size.
Reviewer 3 Report
I've learnt a good deal from this paper, which deals with matters I touch upon in teaching but do not research.
The content is useful, well organised, and presented in almost faultless English. (Note Cambodia spelling).
My only negative comment is the surprising lack of detail in the legend to Figure 3: no origin(s) being given for the data used. I assume the GenBank accession numbers given in Fig 4 are the origin.
Author Response
Authors answer: We appreciate the correction and agree. We added in the legend of Figure 3 what was the origin of the sequences is available in supplementary material.
Round 2
Reviewer 2 Report
The authors have addressed the majority of the concerns.
The figure 1 was very important improvement. However, there is a mistake, in S João do Piauí, with 3 positive and 6 sequenced (impossible)